# What Are the Keys to the Adaptive Success of European Wild Rabbit (*Oryctolagus cuniculus*) in the Iberian Peninsula?

**DOI:** 10.3390/ani11082453

**Published:** 2021-08-20

**Authors:** Pablo Jesús Marín-García, Lola Llobat

**Affiliations:** 1Institute for Animal Science and Technology, Universitat Politècnica de València, Camino de Vera s/n, 46022 Valencia, Spain; 2Department of Animal Production and Health, Veterinary Public Health and Food Science and Technology (PASAPTA), Facultad de Veterinaria, Universidad Cardenal Herrera-CEU, CEU Universities, 46113 Valencia, Spain

**Keywords:** rabbit, alien, endangered, preservation, ecosystem

## Abstract

**Simple Summary:**

Why might a species both be seriously threatened and pose an overpopulation problem in introduced locations? The aim of this review was to understand the keys to the adaptive success of the wild rabbit (*Oryctolagus cuniculus*) in order to establish its strengths and weaknesses for the management of this keystone species in Mediterranean ecosystems. This work highlights the need to create specific conservation programs for this species.

**Abstract:**

The European wild rabbit (*Oryctolagus cuniculus*) plays an important ecological role in the ecosystems of the Iberian Peninsula. Recently, rabbit populations have drastically reduced, so the species is now considered endangered. However, in some places, this animal is considered a pest. This is the conservation paradox of the 21st century: the wild rabbit is both an invasive alien and an endangered native species. The authors of this review aimed to understand the keys to the adaptive success of European rabbits, addressing all aspects of their biology in order to provide the keys to the ecological management of this species. Aspects including nutrition, genetics, immunity interactions with the environment, behaviour, and conflict with human activities were reviewed. Ultimately, rabbits are resilient and adaptable. The main adaptations that explain the rabbit’s adaptive success are its nutrition (wide adaptation to food and good nutritional use of caecotrophy), immune system (powerful and developed), and other aspects related to genetics and behaviour. Rabbits’ relationship with humans has led them to colonise other places where they have become pests. Despite these adaptations, populations in native places have been drastically reduced in recent years. Since it serves as a bastion of the Mediterranean ecosystem, a specific conservation program for this species must be carried out. Therefore, a study of the rabbit’s response to diseases and nutrition (especially protein), as well as the interaction between them, is of special interest.

## 1. Introduction

The European wild rabbit (*Oryctolagus cuniculus*) plays an important ecological role [1] in the ecosystems of the Iberian Peninsula. Rabbits are an essential prey for more than 40 species of carnivores such as *Canis lupus, Aegypius monachus,* and *Strix aluco* [2], and they comprise more than 50% and 75% of the diets of, respectively, the Spanish Imperial Eagle (*Aquila adalberti*) and the Iberian lynx (*Lynx pardinus*) [3], the most threatened feline in the world [4]. The absence of wild rabbits, a key prey species on the Iberian Peninsula, is the most important factor affecting the risk of extinction of both species [5,6]. In the second half of the twentieth century, local extinctions of the wild rabbit began to occur due to the emergence of two diseases: myxomatosis and rabbit haemorrhagic disease (RHD) [7,8]. Villafuerte et al. (1994) estimated a 55% wild rabbit mortality rate in Spain [9]. In the last few years, several national programs have been conducted in Spain to increase the number of wild rabbits and Iberian lynx (ICONA-CSIC, 1988, ICONA, 2005, LIFE02 NAT/E/008609a, b, LIFEþ10 NAT/ES/000570, LIFE06/NAT/E/000209, LIFE92 NAT/E/014300, and LIFE95 NAT/E/004818). In addition, rabbits’ burrows are used as a refuge for many species, such as *Malpolon monspessulanus* and *Bufo bufo* [10,11,12], and their nutritional habits enable the dispersion of important Mediterranean plants [13,14]. For these reasons, rabbits have been defined as a keystone species [15]. Despite its role as a keystone species in its endemic locations, this animal is also considered an invasive species in some places, and great effort has been made to minimise its density or even to make it disappear [16]. Alien species cause damage to the ecosystems of the places where they have been artificially introduced [17,18]. This kind of invasion can lead to large-scale alterations in ecosystem function, and these impacts may include species extinctions [19,20] and even the “meltdown” of ecosystems [21]. Consequently, the control or eradication of invasive alien species is widely undertaken and recommended [22,23]. As an attempt to eradicate rabbits, two diseases were illegally introduced. However, though myxomatosis and rabbit haemorrhagic disease were rapidly spread throughout Europe (even in the species’ endemic areas), the outbreaks failed to remove the introduced rabbits. This, along with habitat deterioration and/or fragmentation, is the main reason for the decimation of European rabbit population [24,25,26] in some Mediterranean areas. As such, rabbits were recently considered to be endangered [27]. The case of the rabbit is peculiar, as it embodies the conservation paradox of the 21st century: it comprises harmful populations in non-native ecosystems and decaying populations in endemic places [28].

The aim of this review was to consider the adaptive success of European rabbits, addressing all aspects of their biology, in order to understand their ecological management, which is necessary in Mediterranean ecosystems.

## 2. Genetics in Rabbits

### 2.1. Origin and Evolution of Oryctolagus cuniculus in Europe

The European rabbit belongs to the Euarchontoglires superorder (formed by the following orders: Rodentia, Scadentia, Dermoptera, Primates, and Lagomorpha [29]). The Lagomorpha first appeared in the late Palaeocene to early Eocene periods of Eurasia and North America [30]. Though its phylogeny is currently not well known, its diversity has decreased from its past ~75 genera and more than 230 species [30]. The Lagomorpha order includes [31,32,33] two families: Ochotonidae (pikas or rock rabbit) and Leporidae. The genus *Oryctolagus* (Leporidae) comprises *O. cuniculus* as the only species with two subspecies—*O. c. cuniculus* and *O. c. algirus* [34]. *O. cuniculus* is considered native to southwestern Europe and Northwest Africa [35,36]. In native areas, rabbits are considered endangered [27], but they can be found on all continents except for Antarctica according to the Global Biodiversity Information Facility (GBIF), where they are considered to be an alien species.

### 2.2. Rabbit Breeds and Genetic Diversity

The wild rabbit was domesticated in the fourth century [37], and more than 100 breeds exist at present [38] (though there is no consensus regarding the number of recognised breeds; the American Rabbit Breeders Association, Taxonomist, and British Rabbit Council recognise 51, 47, and 80 breeds, respectively [39,40,41]); these breeds are distinguished by several traits, such as productive and reproductive [42,43,44], and show clearly relevant phenotypic differences [45] between different synthetic lines. Domestication was probably carried out for a single species and has led to less genetic diversity (around 20% [46]) in domestic rabbits than in wild populations [47]. Genome analysis has shown that domestic rabbits have evolved through many small-effect mutations, unlike other domesticated species [48]. All of this information denotes a relevant genetic plasticity and, consequently, possible adaptive success.

Given that the rabbit has suffered two important viral epizootics (myxomatosis and rabbit haemorrhagic disease) in recent years, one might think that the genetic diversity of the wild rabbit has decreased. However, through the analysis of 14 polymorphic loci, Queney et al. demonstrated that these epizootics had little effect on the genetic diversity of wild rabbit populations, probably because they maintained a sufficiently large population [49]. Though variability between breeds has not been detected, the sequencing analysis of the wild rabbit’s Y gen (SRY) demonstrated that its nucleotide diversity is higher than other species [50]. This genetic diversity affects different polymorphisms of the total genome of rabbit, including plasma proteins and the immune system [51,52,53,54]. This high genetic diversity is probably related to the immune system, which may explain the simultaneous appearance of rabbit populations resistant to myxomatosis in different parts of the world [55].

### 2.3. Immune System of Oryctolagus cuniculus: Genetic Overview

Traditionally, the rabbit has been one of the species most used for immunology studies, which means that many studies on the genetic diversity of the rabbit’s innate and adaptive immune systems have been carried out.

The innate immune system mainly consists of interleukins, chemokines, and other signalling proteins, such as Toll-like receptors and antiviral proteins. Interleukins (ILs) are secreted by leucocytes and could act on other cells. Interestingly, the sequence that codes for the ILs has different loci that evolve under positive selection [56]. Whereas some ILs, such as IL-2 and IL-10, are barely conserved, others, such as IL-4 and Il-6, are slowly conserved at both the nucleotide and amino acid levels [57]. IL-6 has a relevant role in the immune response against rabbit haemorrhagic disease virus (RHDV) [58], and it presents a specific mutation in the stop codon of the European rabbit, which produces a protein with an additional 27 amino acids [57]. Another interleukin required for the differentiation of lymphoid progenitors into B and T cells, IL-7, presents a functional isoform generated by alternative splicing in the European rabbit [59]. Chemokines and their receptors are relevant to immune and inflammatory responses. The genetic diversity between lagomorph genera was studied, and relevant differences in some chemokines and receptors—such as CCR2, CCR5, CCL3, CCL4, CCL5, CCL8, CCL14, and CCL16 (a pseudogene in European rabbit)—have been demonstrated [60,61,62,63,64,65]. However, the possible functional results of these interspecies differences and intraspecies genetic diversity in the European rabbit have not yet been analysed.

Toll-like receptors (TLRs) are type I transmembrane glycoproteins that protect the conserved structures of pathogens (pathogen-associated molecular patterns (PAMPs)) and provoke the immune response [66]. Even though TLRs are conserved in most mammalian species, the European rabbit presents some singularities. For example, in this species, TLR8 presents a low activity, TLR7 is missing [67], and TLR3 (an antiviral TLR that recognises dsRNA) shows a higher diversity pattern than in other rabbit breeds [68], which could be related to resistance against viruses such as myxomatosis and RHDV. Antiviral proteins such as the retinoic acid-inducible gene-I (*RIG-I*) and melanoma differentiation-associated factor 5 (*MDA5*) have shown genetic differences between the European and brush rabbit (*Sylvilagus bachmani*), which could explain the differences in the susceptibility of the myxomatosis virus in the two species [69]. Studies on the genetic diversity of these proteins within European rabbit species should be conducted to determine their ability to genetically adapt to this epizootic.

## 3. Reproduction in Rabbits

*O. cuniculus* is a reproductively efficient species. First, it is a species with induced ovulation, which presents irregular and indefinite oestrus cycles [70]. Unlike the domestic rabbit, which has continuous breeding seasons, the wild rabbit shows clearly seasonal breeding [71,72], probably due to “domestication syndrome” [73]. These reproductive differences between domestic and wild rabbit depend on some candidate genes that regulate clock functions and reproduction [74]. Reproductive activity is mainly determined by factors that mainly affect males, such as their photoperiod (some studies have indicated that testicular activity and growth are affected by circadian cycles [75,76]), and others that affect both females and males, such as social interactions and climatology [77]. However, the most relevant factor related to reproduction seasonality seems to be nutrition. Changes in food supply are particularly relevant for wild rabbits living in highly localised social groups [78]. This reproductive adaptation of the rabbit is worth noting, as the appearance of offspring is avoided in periods with limited food availability, thus leading to low survival rates.

The reproductive efficiency of the rabbit is also determined by other factors. One of the most relevant is the ability of this species to overlap gestation with lactation and, more importantly, efficiency in ovulation and pregnancy. Since few rabbits in oestrus do not produce ovules, the proportion that do not become pregnant is small. Females freely copulate during gestation, pseudopregnancy, and even during the non-breeding season [79]. In addition, the death of entire litters and their subsequent reabsorption cause the animals to enter oestrus again at or shortly after the time they would have if they had been pseudopregnant; the death of some embryos in a polytocal animal, such as the rabbit, rarely interrupts a pregnancy because the dead embryos are reabsorbed without seriously interfering with the development of the rest [79].

In addition to the aforementioned factors that improve the reproductive efficiency of rabbits, one must consider that the rabbit is a social and territorial species, with populations that are subdivided into small independent social groups that occupy defined territories and restrict the arrival of individuals from other groups. The apparent stability of groups during the breeding season suggests their reproductive isolation [80]. However, interbreeding has been observed between social groups, so intragroup social organisation and hierarchy do not necessarily define the genetic makeup of wild rabbit populations [81]. As discussed, nutrition appears to be one of the most important factors related to reproductive success. It would be interesting to investigate how nutrition and genetics are related (epigenetics), because they appear to factor into the rabbit’s environmental adaptation mechanisms [82].

## 4. Nutrition

Only a fraction of the absorbed nutrients of eaten plants is used by the rabbit to improve its productivity (either through growth or reproduction). Any mechanism that improves the proportion of available nutrients for this purpose translates into greater adaptive success.

### 4.1. Availability of Food

The ability to acquire nutrients improves the nutritional input (Figure 1) and maximises the adaptative success of the rabbit. Ferreira and Alves [26] revealed the presence of more than 50 plant species, which varied depending on the season, in the rabbit’s diet in the Mediterranean ecosystem [83]. This capacity confirms the generalist character of this lagomorph. Rabbits explore different vegetation strata and can adapt their feeding strategies to the quantity and quality of the available resources. They have also shown nutritional flexibility in introduced areas [83]. Additionally, the rabbit has developed a strategy for high feed intake (65–80 g kg^−1^ body weight) and the rapid transit of feed through their digestive systems to meet nutritional requirements [84]. In short, rabbits have a great ingestion capacity in terms of both the quantity and types of food.

### 4.2. Alimentation Use Efficacy

*Oryctolagus cuniculus* has an especially peculiar digestive physiology: a powerful caecum (which comprises 50% of the total digestive tract [85], so microbial activity has great importance for the digestion process) and the practice of caecotrophy. Caecotrophy is the ingestion of faeces for nutritional purposes. Rabbits produce two kinds of faeces: soft and hard. Soft faeces are excreted and systematically ingested according to the circadian rhythm (higher intake in the early hours of the day), which is the opposite to that of feed intake and hard faeces excretion (mostly produced in the late hours of the day) [86,87]. Caecotrophy plays an important role in rabbit nutrition by providing proteins and B vitamins from bacteria. Soft faeces contain greater proportions of proteins, minerals, and vitamins than hard faeces, while hard faeces are higher in fibrous components [88]. For this reason, caecotrophy provides an additional source of water, proteins, and vitamins for the rabbit. In farm situations, the contribution of soft faeces to the total CP intake is around 17% [88], though it is predictably higher in the wild. Furthermore, the protein of soft faeces is rich in essential amino acids such as lysine, sulphur, and threonine [89,90].

### 4.3. Nutritional Requirements

There are different maintenance levels for several biological parameters (such as age and reproductive state) but in general, at the laboratory level, rabbits need around 450 KJ day^−1^ kg^−1^ LW0^0.75^ and 2.9 g day^−1^ kg^−1^ LW0^0.75^ of digestible energy and protein, respectively [91,92]. Compared to other highly productive and adaptable species, rabbits allocate (proportionally to their weight) 8% less energy to maintenance [93], which means a greater proportion of nutritional resources is destined for productive or reproductive functions. These low values of maintenance-necessary nutrients could be translated into a greater ability to increase offspring. The rabbit has been shown to efficiently utilise digestible energy for foetal growth and milk production [94,95]. From the nutritional point view, rabbits show formidable maternal aptitudes (females only visit their pups for a few minutes once every 24 h to nurse) [96], showing a large quantity (around 250 g day^−1^) [94,97] and quality of milk (about 2.9 times higher energy than cow milk). The nutritional requirements are defined as the sum of what is needed for maintenance and other productive factors. Meeting nutritional requirements improves adaptive success, which is the main reason for establishing a good methodology to determine them in the wild [98,99,100], as protein supply appears to be the primary population-limiting factor in some ecosystems, and goes towards an ideal protein concept. The nutritional requirements of wild rabbits are summarised in Table 1.

### 4.4. Global Nutritional Assessment

Despite their nutritional adaptations, rabbits have high requirements that apparently cannot be covered in all ecosystems. It has been shown that nutrition (mainly protein) can limit the abundance and density of this species [102], and an increase in food availability (studies of other lagomorphs have shown that a higher crude protein increases the odds that plants are considered for food) [103] has been shown to be a good management technique [25] and optimal for restocking in some areas [101]. This relationship may be explained by a lack of necessary nutrients to achieve a correct reproductive fitness (as previously explained) or by nutrition–health interactions. Rabbit survival is jeopardised under poor nutritional conditions, mainly by increasing predation risk and influencing the susceptibility of individuals to epizootic events. In experimental situations, rabbits are capable of choosing diets with subtly different protein contents [104], and the nutrition/health relationship has been repeatedly investigated [98,105], but how this interaction affects the conservation status of this species is currently unknown. Understanding the nutritional limits of ecosystems and their interactions with diseases will be key in the future management of the species.

## 5. Other Adaptive Success

### 5.1. Behaviour

The rabbit is highly capable of adapting to external changes. These adaptations include many behaviours, such as nursing. Though it was once believed that nursing behaviour followed a circadian rhythm [106], recently, studies have shown otherwise. Rabbits are able to adapt various aspects of their nursing behaviour at each lactation stage, suggesting a breastfeeding pattern regulated by an hourglass-type process with a period of less than 24 h that is restarted with each new lactation [107].

Though it may seem that the presence of more predators or competitors is the main cause of changes in the behaviour of rabbits, the reality is that these changes are more influenced by the quality and quantity of available food. It appears that rabbits can adapt their foraging behaviour and use of space in the wild [108]. However, nutrition in rabbits does not only affect behaviour; it has been demonstrated that rabbits can change their jawbone plasticity to fit the type of ingested food (an example of phenotypic plasticity). These changes have been found in both offspring and adults [109]. Individual environments determine the local selection of plants. Thus, some lagomorph species, such as pygmy rabbits (*Brachylagus idahoensis*) and mountain cottontail rabbits (*Sylvilagus nuttalli*), have been shown to change the amount and composition of food eaten depending on the ambient temperature [110]. These changes have been reflected in modifications of the microbiota. Funosas et al. showed that the intestinal microbiota are highly variable in different populations of wild rabbits, which could signify an adaptation to the environment and its available diet [111].

Human presence is one of the most relevant changes that an environment undergoes. For example, construction projects could modify ecological and environmental architecture. These ecosystem changes affect all present species, each with its own adaptation ability depending on its resilience capacity. Specifically, the rabbit has the capacity to adapt better than many other species, even taking advantage of artificial transport systems such as roads and railways to spread [112].

### 5.2. Environment

The Mediterranean climate is clearly characterised by mild winters and hot and dry summers [113]. These summers push biotic individuals to their limits, so rabbits have relevant adaptations [114]. For instance, the rabbit’s large ears enable thermoregulation by dissipating high temperatures [115]. In addition, caecotrophy provides an extra supply of water (as mentioned in the nutrition section). Rabbits can build their own burrows, which allows for a more controlled environment, and their faeces provide a substrate to the soil that enriches the appearance of plant species. These adaptations allow for survival in unfavourable situations. A study carried out in the Canary Islands showed that the presence of rabbits intensified with increasing temperatures and reducing rainfall, thus demonstrating this adaptive potential [116]. In general, the rabbit is clearly adapted to its native ecosystems.

### 5.3. Human Relationship

Rabbits are a species that has been widely hunted [117] because it is prized as small game. This has been one of the main reasons for its introduction to many different ecosystems. Additionally, the rabbit has a significant relationship with humans [118]; it is widely used as a domestic and livestock animal (for meat and other productive traits) in many countries, such as the United States and the British Islands. Another example of human/rabbit interaction is the animals’ use of roads and railway lines as potential corridors [112]. In addition, the rabbit’s adaptability allows it to find food in anthropogenic places such as cultivated areas [119], and, in locations with favourable conditions, rabbits conditioned to the existence of natural predators and their associated problems can produce large population increases. Overall, the rabbit’s strong tolerance to humans has provided certain adaptive benefits.

## 6. Overview

Rabbit populations have been greatly reduced to the point of being seriously threatened in recent years. Rabbits have a bad reputation in some areas because they conflict with human beings [120]. This conflict has led to the thinking that they exist in excess, which has hindered the creation of conservation awareness. In this review, we intended to convey the importance of this species beyond its sustenance for any other animal. Many of the beneficial effects of this species do not lie in its presence/absence; rather, the effects lie in the allowance of a large population that enables ecosystem sustenance. For example, it has been demonstrated that the Iberian lynx needs an abundance of rabbits [121], which suggests the need to vary the conservation and protection efforts of this species based on the needs of specific populations. The loss and degradation of the forests where the rabbit lives, as well as the fragmentation of ecosystems, could produce an unstoppable population decline that we must stop to stabilise its populations.

Rabbits are a resilient animal. They have shown powerful adaptations in terms of nutrition, genetics, and immunity, among others, that explain their invasive potential. We understand the potential effects of these adaptations in a typically Mediterranean climate. Caecotrophy allows for rehydration and a better use of proteins. Studies have suggested that the maximum limitations of these populations are the presence of introduced diseases and protein availability. Therefore, for their conservation, it will be especially interesting to study the rabbit’s nutrition and immune system, as well as the possible interactions between them, and approaches that also integrate production could lead to viable strategies for its protection and conservation. These strategies could affect more than just this species; they may be applicable to others such as *Bunolagus monticularis,* another critically endangered lagomorph [122].

From the information presented in this review, two main conclusions can be drawn. First, the rabbit is a species that must be protected and conserved, since it is a keystone species in fragile ecosystems. Secondly, considering human/wildlife interactions will be important for the continued survival of the rabbit and other species.

## 7. Conclusions

Rabbits are resilient and adaptable animals. The main adaptations that explain their adaptive success are those regarding their nutrition (great variety of choice and high use), immune system (powerful and developed), and others related to genetics and behaviour. Their relationship with humans has led them to colonise other places where they have become pests (while making use of their adaptations). However, in the places where they are autochthons, their populations have been drastically reduced in recent years. Since the rabbit is a bastion of the Mediterranean ecosystem, a targeted conservation program for this species must be carried out. A study of its responses to diseases and nutrition (especially proteins), as well as the interaction between them, should be considered as especially worthy of interest.

## Figures and Tables

**Figure 1 animals-11-02453-f001:**
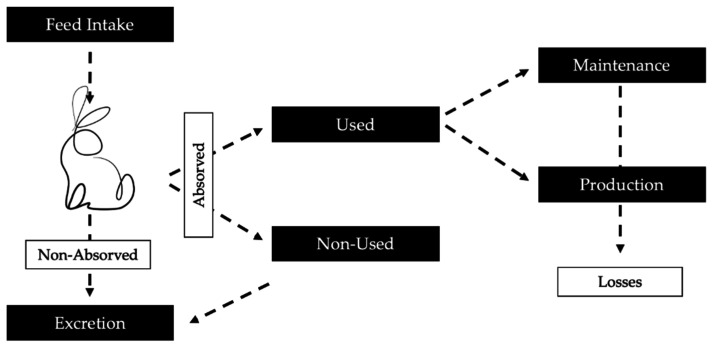
Nutritional distribution within an animal.

**Table 1 animals-11-02453-t001:** Nutritional requirements of rabbits obtained from different studies [89,101].

Physiological Process	Nutritional Requirements
Digestible energy requirement for maintenance	450 KJ day^−1^ kg^−1^ LW0^0.75^
Digestible protein requirement for maintenance	2.9 g day^−1^ kg^−1^ LW0^0.75^
Milk production	250 g day^−1^
Lysine requirement for breeding	8.0 g Kg^−1^
Sulphur amino acid requirement for breeding	6.3 g Kg^−1^
Threonine requirement for breeding	6.4 g Kg^−1^
Digestible energy	10.51 MJ Kg^−1^

## Data Availability

Not applicable.

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
