# Peer review of "What Are the Keys to the Adaptive Success of European Wild Rabbit (Oryctolagus cuniculus) in the Iberian Peninsula?"

_animals, 2021, doi:10.3390/ani11082453_

Round 1

Reviewer 1 Report

The authors carry out a  review to detect the keys to the adaptive success of the European rabbit. Review is ambitious and could provide relevant information on the subject. The ability of the rabbit to adapt to different ecosystems is debated, mainly due to nutrition and its immune system as well as genetics, behavior and reproduction.

But, in my opinion the scientific level of the review is poor. The title does not agree with the development of the review. An example, the authors have focused on the Iberian peninsula (line 36-37) although the wild rabbit is found all over the world (except in Antarctica). Or "conservation consequences" is indicated but little reference is made in the review.
The introduction is erratic and leads to a paradox (lines 60-65) but then the objective of the review is to detect the keys to the adaptive success of the European rabbit.
On some occasions, terms are explained and it is not necessary to do so for a specialized reader (for example: lines 181-187; 217-220, etc.)
The figures presented are  simple and do not provide interesting information to the text. However, tables with data that support many of the statements made by the authors throughout the text are missing.

Authors abuse parentheses and this makes the text difficult to read.
Some other considerations:
line 65: what is the purpose of this question?

line 152-153. Reproductive variations are only justified by the photoperiod via the male, and the female ?

Line 163-164.What do the authors want to indicate with this sentence?
Overview: it is mainly a summary of the previous points. But the last parragraphs have to be reconsidered. No data has been provided on animals to be protected and conserved or the purpose of this conservation. In addition, no information has been provided for the assertion of lines 346-348.

Therefore, I consider that the manuscript is not appropriate for publication in  Animal

Author Response

Dear reviewer,

Thank you very much for all the corrections and recommendations that you have made. We agree with all the annotations that you have made, and we have modified them all. We would like to thank the effort made as we are interested in improving our work. Next, we respond point by point to them:

The title does not agree with the development of the review. An example, the authors have focused on the Iberian Peninsula (line 36-37) although the wild rabbit is found all over the world (except in Antarctica). Or "conservation consequences" is indicated but little reference is made in the review.

Indeed, our review is focused on those areas where the wild rabbit is a threatened species, since this species presents this duality of threat species in some areas and invasive species in others (for more information, the reviewer can consult the article from 2008 de Lees and Bee, DOI: 10.1111 / j.1365-2907.2008.00116.x). In order to clarify this point, we have modified the title of the review as “What are the keys to the adaptative success of European Wild Rabbit (Oryctolagus cuniculus)?

The introduction is erratic and leads to a paradox (lines 60-65) but then the objective of the review is to detect the keys to the adaptive success of the European rabbit.

We agree that the introduction leads to a paradox. This is precisely the focus of the review. To make this point better understood, we have modified this part of the introduction.

On some occasions, terms are explained, and it is not necessary to do so for a specialized reader (for example: lines 181-187; 217-220, etc.).

Thank you very much for the recommendation. We agree that the statements that the reviewer has indicated are obvious. For this reason, these phrases have been removed from the manuscript. The figures presented are simple and do not provide interesting information to the text. However, tables with data that support many of the statements made by the authors throughout the text are missing. 

Thank you very much for your recommendation. Following that, we have removed one figure and we have added another table with data that support the information of the manuscript.

Authors abuse parentheses and this makes the text difficult to read.

Done. we have eliminated the use of parentheses and better organized the text.

Some other considerations:

line 65: what is the purpose of this question?

Thank you for your recommendation. Following your instructions, the question has been removed from the manuscript to avoid confusion in understanding

line 152-153. Reproductive variations are only justified by the photoperiod via the male, and the female?

Since the rabbit is a species of induced ovulation, that is, that the females of this species do not have heat, the variability in reproduction is regulated mainly by the reproductive capacity of the males. However, the reproduction of females is affected that social interactions and climatology, among others. This information has been added in the manuscript.

Line 163-164. What do the authors want to indicate with this sentence?

In this sentence, we refer to the fact that, unlike other species, the female accepts copulation not only in heat, but also in other periods, even when she is pregnant and lactating. We have modified the sentence to improve your understanding.

Overview: it is mainly a summary of the previous points. But the last paragraphs have to be reconsidered. No data has been provided on animals to be protected and conserved or the purpose of this conservation. In addition, no information has been provided for the assertion of lines 346-348.?

Thank you very much for your recommendation. We changed the last sentences according to the reviewer and we have added more information, including data about number of animals to be protected and conserved..

Reviewer 2 Report

Line 14 : keystone species

Line 23: It is preferable to say: “interaction with the environment, behaviour, conflict with human activities, among others…”

Line 24 : “In overall words, rabbits are resilient and very adaptable animals.”

Line 45: “keystone species.”

Line 53: “As an attempt to eradicate rabbits, several viruses 53 were introduced illegally.” Add a reference or remove. Several gives the idea that were many many viruses, in fact, were at maximum 2, Additionally the introduction of the viruses was in some cases authorized as biological control.

Line 54 - “However, the outbreak of myxomatosis and rabbit hemorrhagic disease failed to remove the introduced rabbits (example Australia), and these viruses were spread rapidly throughout Europe (even in its endemic places).”. Please play attention that myxomatosis and RHD are diseases and then refered as “these viruses”.

Line 72 – “Phylogenetic evolution of wild rabbit is shown in Figure 1.” I do not agree that figure 1 shows a phylogenetic evolution at all. The figure is too simplified and has low quality figures. For example, primates and lagomorpha / rodents were known to have an ancestor in common and that these two groups diverged. Phylogenetic evolution must have a scale of time and genetic distance that is not at all demonstrated here. I suggest removing the figure or greatly increase its complexity and scientific reality.3

Line 79 : Use species and not specie

Line 80 “O. cuniculus is considered native to south-western Europe and north-west Africa [33,34] were are considered as endan-81 gered [22].” This sentence is not clear

Line 87 : “4th century” bc or ac? Exist not exits

Line 99 “viral hemorrhagic disease of the rabbit”, the oficial disease name is rabbit hemorrhagic disease or RHD.

Line 117 “whereas some IL, as IL-2 and IL-10 are highly conserved, others as IL-4 and Il-6 are 117 strongly conserved, at both nucleotide and amino acid levels [56].” Highly or strongly are not the same?

Line 127 “CCL3, CCL4, CCL5, CCL8. CCL14, and CCL16,”. The final point between CCL8 and CLL14 should not exist, right?

The manuscript may be extensive revised in the english to allow the reviewing. It becomes very tiring to be correcting these small mistakes

Author Response

Dear reviewer,

Thank you very much for all the corrections and recommendations that you have made. We agree with all the annotations that you have made, and we have modified them all. We would like to thank the effort made as we are interested in improving our work. Next, we respond point by point to them:

Line 14 : keystone species.

Done.

Line 23: It is preferable to say: “interaction with the environment, behaviour, conflict with human activities, among others…”

Done.

Line 24 : “In overall words, rabbits are resilient and very adaptable animals.”

Done.

Line 45: “keystone species.”

Done.

Line 53: “As an attempt to eradicate rabbits, several viruses 53 were introduced illegally.” Add a reference or remove. Several gives the idea that were many many viruses, in fact, were at maximum 2, Additionally the introduction of the viruses was in some cases authorized as biological control.

Done.

Line 54 - “However, the outbreak of myxomatosis and rabbit hemorrhagic disease failed to remove the introduced rabbits (example Australia), and these viruses were spread rapidly throughout Europe (even in its endemic places).”. Please play attention that myxomatosis and RHD are diseases and then refered as “these viruses”.

Done.

Line 72 – “Phylogenetic evolution of wild rabbit is shown in Figure 1.” I do not agree that figure 1 shows a phylogenetic evolution at all. The figure is too simplified and has low quality figures. For example, primates and lagomorpha / rodents were known to have an ancestor in common and that these two groups diverged. Phylogenetic evolution must have a scale of time and genetic distance that is not at all demonstrated here. I suggest removing the figure or greatly increase its complexity and scientific reality.

Done.

Line 79 : Use species and not specie

Done.

Line 80 “O. cuniculus is considered native to south-western Europe and north-west Africa [33,34] were are considered as endan-81 gered [22].” This sentence is not clear

Done.

Line 87 : “4th century” bc or ac? Exist not exits

Done.

Line 99 “viral hemorrhagic disease of the rabbit”, the oficial disease name is rabbit hemorrhagic disease or RHD.

Done.

Line 117 “whereas some IL, as IL-2 and IL-10 are highly conserved, others as IL-4 and Il-6 are 117 strongly conserved, at both nucleotide and amino acid levels [56].” Highly or strongly are not the same?

Done.

Line 127 “CCL3, CCL4, CCL5, CCL8. CCL14, and CCL16,”. The final point between CCL8 and CLL14 should not exist, right?

Done.

The manuscript may be extensive revised in the english to allow the reviewing. It becomes very tiring to be correcting these small mistakes.

We regret the difficulty found by the reviewer to review the manuscript due to the English. Since no reviewer has previously instructed us that we should send it to a language reviewer, we have not. We also know that "Animals" magazine reviews English carefully before publishing the papers. However, if the journal and the reviewer consider that the manuscript should be thoroughly revised with respect to English, we have no problem.

Round 2

Reviewer 1 Report

The authors have considered some of the proposed indications. But I am still considering the following aspects to improve:

1. The authors still do not justify with data the number of animals (breeds) threatened or the justification for conservation.
2. In the title they have indicated that they focus on the European wild rabbit, but the first sentence of the introduction focuses on the Iberian Peninsula. There are other areas, not only in southern Europe but also in North Africa with a high population of rabbits.
Furthermore, the manuscript has a very low quality presentation. Although I am not a specialist in English, I believe that the text has grammatical errors and a lack of the correct expression. Also the bibliography is not well cited, for example citations from 24 to 26 are absent.

Lines 68-78. Please, review english. Up to 5 times "current" is written in the paragraph.

Lines 251-252: Please rewrite.

Line 321: Please rewrite

Author Response

The authors have considered some of the proposed indications. But I am still considering the following aspects to improve:

  1. The authors still do not justify with data the number of animals (breeds) threatened or the justification for conservation.

Thank you very much for your recommendation. The data related to number of species that depends of wild rabbit, threatened species and justification for conservation have been added in the manuscript (lines 37-53).

  1. In the title they have indicated that they focus on the European wild rabbit, but the first sentence of the introduction focuses on the Iberian Peninsula. There are other areas, not only in southern Europe but also in North Africa with a high population of rabbits.

Indeed, our interest is focused on the conservation of the European wild rabbit in the Iberian Peninsula. To improve the understanding of the manuscript, we have modified the title as " What Are the Keys to the Adaptive Success of European Wild Rabbit (Oryctolagus cuniculus) in the Iberian Peninsula?

Furthermore, the manuscript has a very low quality presentation. Although I am not a specialist in English, I believe that the text has grammatical errors and a lack of the correct expression. Also the bibliography is not well cited, for example citations from 24 to 26 are absent.

Sorry for our mistake. The references have been checked again and the mistake has been corrected. Related to English, the manuscript has been sent to the journal's English reviewers and we enclose the certificate.

Lines 68-78. Please, review english. Up to 5 times "current" is written in the paragraph.

Done.

Lines 251-252: Please rewrite.

Done.

Line 321: Please rewrite

Done.

Reviewer 2 Report

The authors did not revise the English as requested. The writing is really bad, for example in the Acknowledgments : "We are grateful to Veterinary Medicine Faculty to Universidad Cardenal Her- 357
rera, CEU."

Author Response

The authors did not revise the English as requested. The writing is really bad, for example in the Acknowledgments : "We are grateful to Veterinary Medicine Faculty to Universidad Cardenal Herrera, CEU."

Thank you very much for your recommendation. The manuscript has been sent to the journal's English reviewers and we enclose the certificate.

Round 3

Reviewer 1 Report

The authors have adequately considered the suggested recommendations

Author Response

Thank you very much for your comments. 

Reviewer 2 Report

The manuscript has improved a lot, there is no comparison. Now it's a pleasure to read.

Line 41 – both diseases emerged in the second half (around 1950 and 1988, respectively for MYXV and RHDV).

Line 43 and elsewhere – “viral haemorrhagic disease (VHC)”, Please change the name to “Rabbit haemorrhagic disease (RHD)” that is the official name.

Author Response

Thank you very much for your comments. The authors have made the suggested changes:

Line 41 – both diseases emerged in the second half (around 1950 and 1988, respectively for MYXV and RHDV). Done

Line 43 and elsewhere – “viral haemorrhagic disease (VHC)”, Please change the name to “Rabbit haemorrhagic disease (RHD)” that is the official name. Done